

# Move it or lose it: interspecific variation in risk response of pond-breeding anurans

Philip Matich[1,2] and Christopher M. Schalk[3]

[1] Department of Marine Biology, Texas A&M University—Galveston, Galveston, TX, USA
[2] Texas Research Institute for Environmental Studies, Sam Houston State University, Huntsville, TX, USA
[3] Arthur Temple College of Forestry and Agriculture, Stephen F. Austin State University, Nacogdoches, TX, USA

## ABSTRACT

Changes in behavior are often the proximate response of animals to human disturbance, with variability in tolerance levels leading some species to exhibit striking shifts in life history, fitness, and/or survival. Thus, elucidating the effects of disturbance on animal behavior, and how this varies among taxonomically similar species with inherently different behaviors and life histories is of value for management and conservation. We evaluated the risk response of three anuran species—southern leopard frog (*Lithobates sphenocephalus*), Blanchard's cricket frog (*Acris blanchardi*), and green tree frog (*Hyla cinerea*)—to determine how differences in microhabitat use (arboreal vs ground-dwelling) and body size (small vs medium) may play a role in response to a potential threat within a human-altered subtropical forest. Each species responded to risk with both flight and freeze behaviors, however, behaviors were species- and context-specific. As distance to cover increased, southern leopard frogs increased freezing behavior, green tree frogs decreased freezing behavior, and Blanchard's cricket frogs increased flight response. The propensity of green tree frogs to use the canopy of vegetation as refugia, and the small body size of Blanchard's cricket frogs likely led to greater flight response as distance to cover increased, whereas innate reliance on camouflage among southern leopard frogs may place them at greater risk to landscaping, agricultural, and transportation practices in open terrain. As such, arboreal and small-bodied species may inherently be better suited in human altered-landscapes compared to larger, ground-dwelling species. As land-use change continues to modify habitats, understanding how species respond to changes in their environment continues to be of importance, particularly in ecosystems where human-wildlife interactions are expected to increase in frequency.

Corresponding author
Philip Matich, pmatich@tamug.edu

## INTRODUCTION

Natural and anthropogenic disturbances continue to alter populations across terrestrial, freshwater, and marine ecosystems (*Abatzoglou & Williams, 2016*; *Hughes et al., 2017*; *Pecl et al., 2017*). Often as a first response to perturbation, changes in behavior

precede shifts in life history, fitness, and survival, which are of considerable interest for conservation in light of many imperiled species (*Wong & Candolin, 2015*). Habitat and dietary generalists are inherently more adept at responding to disturbance (*Devictor, Julliard & Jiguet, 2008*; *Hamer & McDonnell, 2008*; *Clavel, Julliard & Devictor, 2011*), however, species vary due to interspecific variability in morphology, physiology, and innate behavioral characteristics, regardless of being a specialist or generalist (*McKinney, 2006*; *Battisti, Poeta & Fanelli, 2016*; *Legrand et al., 2017*). As such, understanding how perturbations lead to changes in behavior, and how this may vary across taxonomic groups is important for developing a robust ecological paradigm for disturbance ecology (*Caruso et al., 2016*; *Lany et al., 2017*; *Liu et al., 2017*).

A variety of species have adjusted their behavior accordingly in response to a growing human footprint (*Bateman & Fleming, 2014*; *Wong & Candolin, 2015*; *Caruso et al., 2016*). For example, many birds and mammals have progressively increased their capacity to use human-altered habitats, and even thrive in some suburban and urban areas (*Chace & Walsh, 2006*; *Bateman & Fleming, 2012*; *Meillère et al., 2015*; *Luscier, 2018*). Similarly, other species have significantly increased in population sizes in response to increased food availability and lower predation risk in agricultural areas and parks (*Ganzhorn & Abraham, 1991*; *Naughton-Treves, 1998*). Not all species, however, have responded positively to anthropogenic impacts. Habitat alterations have been detrimental in many ecosystems, particularly for species that are less mobile and/or have stricter ecological and physiological requirements, like amphibians (*Hamer & McDonnell, 2008*; *Crump, 2009*; *Hughes et al., 2017*; *Tilman et al., 2017*). Compared to endotherms, amphibians have limited mobility due to physiological constraints, which results in small home ranges (*Wells, 2007*). Reduced mobility decreases the ability of amphibians to leave altered or degraded habitats, and our understanding of the proximate responses of amphibians (i.e., behavior) to land-use change is critical to inform conservation strategies focused on the long-term persistence of amphibian populations (*Pechmann & Wilbur, 1994*; *Gibbs, 1998*; *Houlahan et al., 2000*; *Crump, 2009*).

Amphibians are among the most threatened taxonomic groups worldwide, with fungal infections and habitat degradation serving as leading causes of species declines (*Stuart et al., 2004*; *Wake & Vredenburg, 2008*; *Grant et al., 2016*; *Lips, 2016*). Approximately one third of amphibian species are threatened or endangered (IUCN; *Collins & Storfer, 2003*; *Stuart et al., 2004*), highlighting the urgency of improving our understanding of their behavioral responses to perturbations, and their capacity to keep pace with human actions (*Arroyo-Rodriguez et al., 2017*; *Lourenco et al., 2017*; *Tilman et al., 2017*). Of concern beyond habitat availability is the response of amphibians to human-induced risk in altered habitats, because of the small body size and primary antipredator response of many species—camouflage. Remaining stationary is an effective antipredator response among many amphibians for natural predators that rely on visual cues (*Crowshaw, 2005*; *Ioannou & Krause, 2009*; *Stevens & Merilaita, 2011*; *Dodd, 2013*; *Bulbert et al., 2017*). However, remaining stationary may be detrimental in response to other risks, such as humans walking or driving vehicles that could lead to non-consumptive mortality

(i.e., being stepped on or run over; sensu *Andrews & Gibbons, 2005*; *Beebee, 2013*; *Heigl et al., 2017*).

Here, we investigate the response of three anuran species with different body sizes, levels of mobility, and microhabitat use patterns—southern leopard frog (*Lithobates sphenocephalus*), Blanchard's cricket frog (*Acris blanchardi*), and green tree frog (*Hyla cinerea*)—to perceived risk at altered edge habitats in a subtropical forest, in order to improve our understanding of the effects of habitat alterations on amphibian response to perturbation. We predicted that in non-vegetated edge habitats, the arboreal species (green tree frog) would exhibit a greater risk response (i.e., flight) than the other species based on its propensity to be found in the canopy of vegetation (*Dodd, 2013*). We also predicted that Blanchard's cricket frogs would exhibit a greater risk response than southern leopard frogs, because of their smaller body size (*Dodd, 2013*).

## MATERIALS AND METHODS

### Study site and species

Our study took place at Sam Houston State University's Center for Biological Field Studies (N30°45′ W95°25′), which is bordered by the Sam Houston National Forest to the south and east, and private ranching and timber holdings to the north and west. Within the Center for Biological Field Studies, pine-hardwood forest, open prairie, old-field succession, and riparian zones are the most abundant habitats (*Dent & Lutterschmidt, 2001*), with amphibian monitoring conducted around ephemeral and permanent ponds.

Amphibians within the study area are locally abundant across much of their range and use a variety of altered and unaltered habitats (*Pyburn, 1958*; *Dodd et al., 2007*; *Dodd, 2013*), making them good model organisms for arboreal species (green tree frog) and semi-aquatic species (southern leopard frogs and Blanchard's cricket frogs) spanning a range of small (Blanchard's cricket frog) to medium body sizes (southern leopard frog). While all three species breed in lentic aquatic habitats, Blanchard's cricket frogs and green tree frogs breed in permanent aquatic habitats, while southern leopard frogs breed in both temporary and permanent ponds. All three species are primarily generalized invertivores that utilize a sit-and-wait foraging strategy (*Dodd, 2013*). However, southern leopard frogs are also capable of consuming small vertebrates (e.g., fishes, other frogs; *Dodd, 2013*). Southern leopard frogs call year-round in East Texas, while calling activity is concentrated between April and August for both Blanchard's cricket frogs and green tree frogs (*Saenz et al., 2006*).

When immobile, the coloration patterns of Blanchard's cricket frogs make them cryptic on land, however, they are strong jumpers that can quickly change direction when approached (*Dodd, 2013*). Cricket frogs are also known to conceal themselves in vegetation. As such, Blanchard's cricket frogs may respond to an approaching threat initially through crypsis, and then seek vegetative cover. The match between green tree frogs and surrounding vegetation coupled with their reduced diurnal movement serves as the primary anti-predator strategy for green tree frogs when in the canopy of vegetation (*Dodd, 2013*). However, when disturbed, individuals are capable of leaping long distances to evade predators (*John-Alder, Morin & Lawler, 1988*). Southern leopard frogs utilize

multiple anti-predator strategies, and can remain motionless and lower their body in a crouching position, or conceal themselves in vegetation (*Marchisin & Anderson, 1978*). The dorsal color and spot patterning enable individuals to match background vegetation (*Dodd, 2013*). Similar to Blanchard's cricket frogs, leopard frogs likely respond first through crypsis, and then seek vegetative cover. Southern leopard frogs also emit a warning scream that functions to startle approaching predators, enabling individuals to escape.

## Data collection

Four, *ca.* 1.5 m-wide belt transects (*Amo, Lopez & Martin, 2006*) 250, 500, 750, and 750 m long were walked at or after dusk, four times monthly from June to November 2017 to conduct visual surveys ($n$ = 96). Transects were paths and hatchery pond berms where vegetation had been cleared through mowing at the edges of forested habitat. The same belt transects were evaluated for each sampling event, in a randomly selected order, beginning <0.01–3.78 h after sunset, depending on selection order (mean = 1.06 h after sunset ± 0.70 SD). Transects were divided into three sections—middle, edge, and boundary—to evaluate how the risk response of frogs to a potential threat (presence of a human) was influenced by proximity to cover/vegetation (Fig. 1). Middle sections were located within the interiors of belt transects, *ca.* 25 cm from cover with limited vegetation (i.e., low cut grass); edge sections were adjacent to middle section *ca.* 0–25 cm from cover with limited vegetation (i.e., low cut grass); and boundary sections were adjacent to edge sections at the boundary of transects, *ca.* 25 cm into vegetation (grasses, forbes, trees).

The same researcher walked slowly on each transect (<1 m/s) to detect anurans, and then moved more slowly (<0.5 m/s) upon encounters to simulate a threat, but effectively identify anurans (*Cooper, 2009*). Once detected, study taxa were visually identified to species, location on transect was recorded (middle, edge, or boundary), and risk response was recorded—stationary (S; i.e., a freeze response), mobile (M—moving upon encounter; i.e., a flight response), or mobile-then-stationary (MS—mobile, then stationary after initial movement; i.e., a flight-then-freeze response). Research was conducted under Sam Houston State University IACUC #17-02-13-1034-3-01.

## Data analysis

Data were organized by individual animal encounter. A generalized linear model using logistic regression was used to investigate the effects of species and transect section (middle, edge, boundary) on frog risk response. Due to the seasonal variability in environmental conditions within the subtropical study location, and habitat use patterns of the study species (*Dodd, 2013*), monthly variability in frog behavior was considered as a predictor variable in addition to species and transect section. Friedman's test was used to investigate the potential effects of sampling across different months, with significant results ($\chi^2$ = 35.56, $p$ < 0.01). Therefore, sampling month was included as a factor in the model. Significant differences were not found across transects ($\chi^2$ = 1.06, $p$ = 0.90), thus data were pooled. All main effects (species, section, month) and two-way interactions were included in the model:

$$H_{ijkl} = \mu + s_j + x_k + m_l + (sx)_{jk} + (sm)_{jl} + (xm)_{kl} + \varepsilon_{ijkl} \tag{1}$$
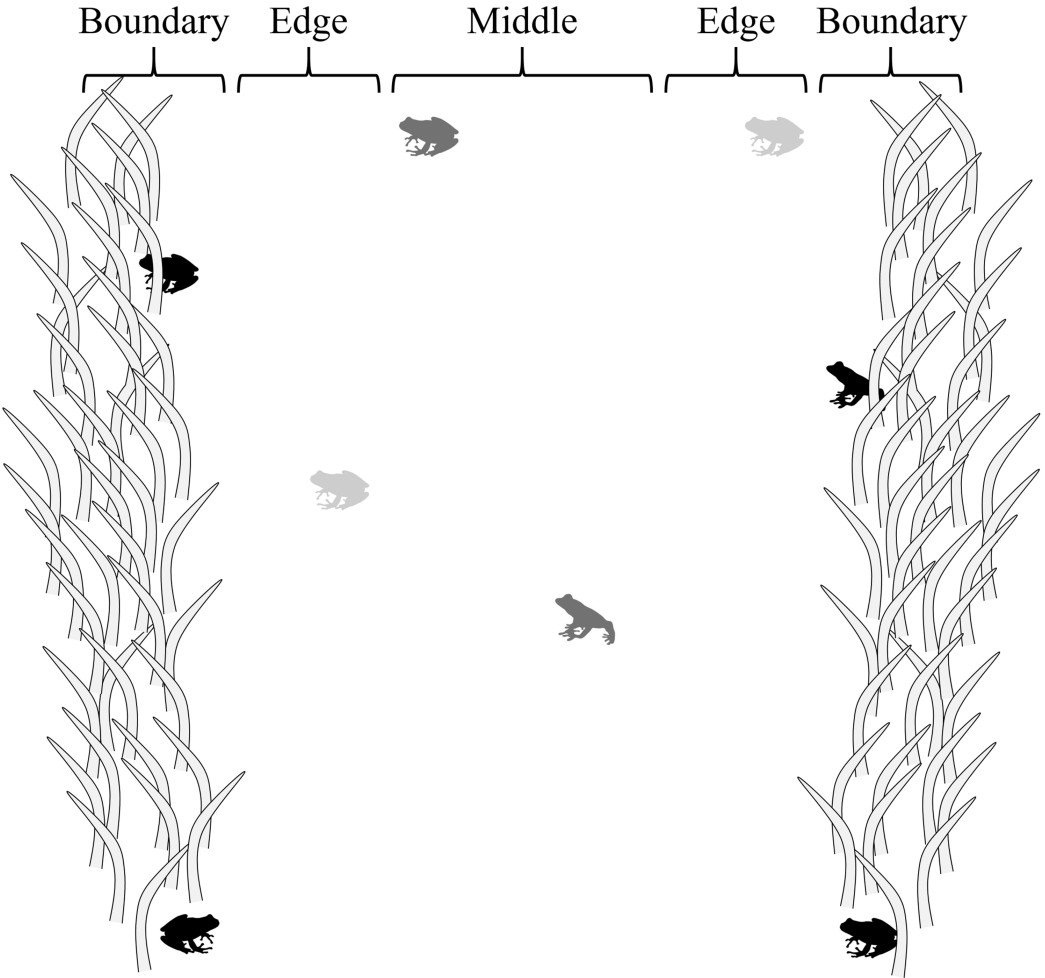

**Figure 1 Transect sections used to evaluate how the risk response of frogs was influenced by proximity to cover.** Middle sections were located *ca.* 25 cm from cover; edge sections were *ca.* 0–25 cm from cover; and boundary sections were adjacent to edge sections at the boundary of transects, *ca.* 25 cm into vegetation. In the figure, black frogs are in boundary sections, light gray frogs are in edge sections, and dark gray frogs are in the middle section.               

Where $s$ is species, $x$ is transect section, $m$ is month, $i$ is the number of sampling events, $j$ is the number of species, $k$ is the number of sections, and $l$ is the number of months. Significance thresholds were corrected for multiple post hoc comparisons. All analyses were conducted in IBM SPSS 22.

## RESULTS

From June to November 2017, 639 southern leopard frogs, 247 Blanchard's cricket frogs, and 1,800 green tree frogs were encountered during 24 nights of sampling across four belt transects ($n = 96$ total; Table 1). Leopard frogs and cricket frogs were more abundant in September–November, while green tree frogs were more abundant in July–September (Table 1).

**Table 1 Sample sizes of study species observed during sampling period.**

| Species | Month | N |
|---|---|---|
| *L. sphenocephalus* | June | 54 |
| | July | 46 |
| | August | 36 |
| | September | 170 |
| | October | 163 |
| | November | 170 |
| *A. blanchardi* | June | 18 |
| | July | 13 |
| | August | 39 |
| | September | 45 |
| | October | 30 |
| | November | 102 |
| *H. cinerea* | June | 166 |
| | July | 777 |
| | August | 340 |
| | September | 268 |
| | October | 151 |
| | November | 98 |

**Note:**
 Sampling period was from June to November 2017.

Among southern leopard frogs, 217 encounters were in the middle of transects (34%), 179 were on transect edges (28%), and 243 were on transect boundaries (38%). A total of 66 Blanchard's cricket frogs were encountered in the middle of transects (27%), 65 were on transect edges (26%), and 116 were on transect boundaries (47%). Among green tree frogs, 42 encounters were in the middle of transects (2%), 56 were on transect edges (3%), and 1,702 were on transect boundaries (95%).

The generalized linear model ($\chi^2 = 563.88$, $df = 73$, $p < 0.01$) indicated that species, transect section, and sampling month, as well as each two-way interaction were significant predictors of frog behavior (Table 2; Table S1). Evaluation against an intercept-only model indicated the significance of the model ($\chi^2 = 2,029.99$, $df = 33$, $p < 0.01$). Green tree frogs exhibited greater stationary behavior than Blanchard's cricket frogs, which exhibited more mobile-then-stationary behavior ($\chi^2 = 1,482.98$, $p < 0.01$). Green tree frogs also exhibited greater stationary behavior than southern leopard frogs compared to mobile ($\chi^2 = 282.21$, $p < 0.01$) and mobile-then-stationary behavior ($\chi^2 = 1,325.34$, $p < 0.01$). Blanchard's cricket frogs exhibited more mobile-then-stationary behavior than southern leopard frogs ($\chi^2 = 30.04$, $p < 0.01$).

In general, frogs were more mobile at transect edges ($\chi^2 = 6.13$, $p = 0.01$) and middles ($\chi^2 = 14.86$, $p < 0.01$) than boundaries, however, this varied across species ($\chi^2 = 92.45$, $p < 0.01$). As distance from vegetation increased (i.e., boundary to edge to middle), southern leopard frogs increased stationary behavior ($\chi^2 = 101.68$, $p < 0.01$), switching from mobile and mobile-then-stationary behavior at transect boundaries,
**Table 2 Test statistics for the generalized linear model.**

| Factor | $\chi^2$ | df | p-value |
|---|---|---|---|
| Species | 139.59 | 2 | <0.01 |
| Transect section | 10.71 | 2 | <0.01 |
| Month | 24.57 | 5 | <0.01 |
| Species*Section | 140.64 | 4 | <0.01 |
| Species*Month | 31.76 | 10 | <0.01 |
| Section*Month | 33.06 | 10 | <0.01 |

Notes:
Test statistics for the generalized linear model investigating the effects of species, location on transection (i.e., section), and sampling month on the risk response of *A. blanchardi*, *H. cinerea*, and *L. sphenocephalus*.

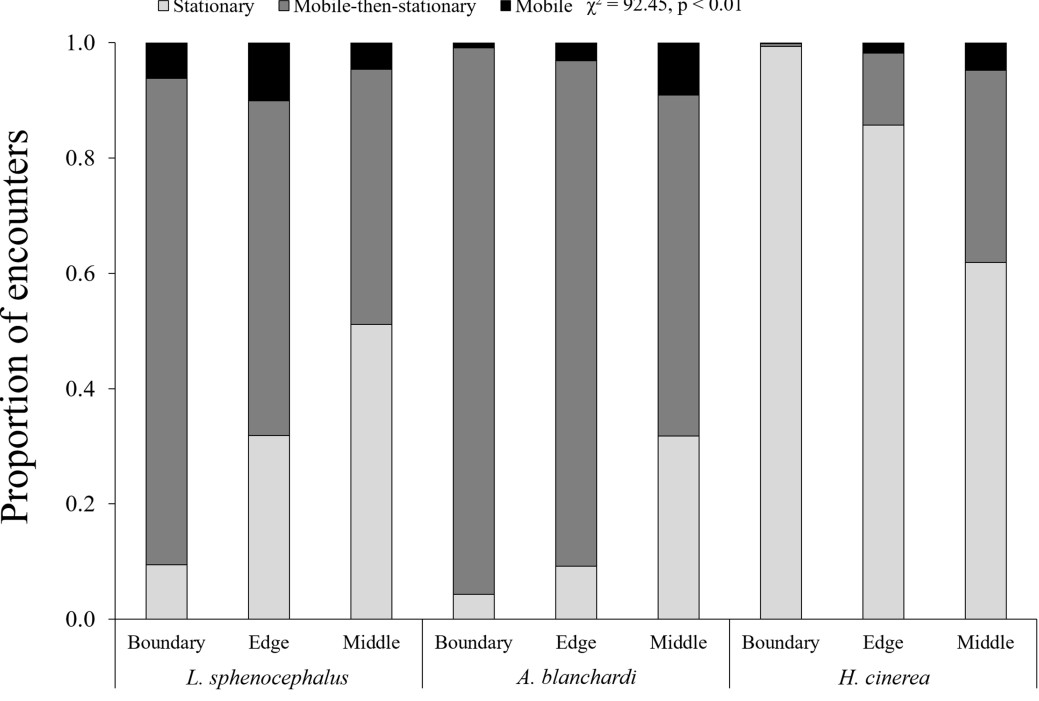

**Figure 2 Spatial patterns in frog behavior.** Proportion of encounters among study species in transect sections that remained stationary (S; light gray), mobile-then-stationary (MS; dark gray), and mobile (M; black) among southern leopard frogs (*L. sphenocephalus*), Blanchard's cricket frogs (*A. blanchardi*), and green tree frogs (*H. cinerea*).

to mobile-then-stationary and stationary behavior at transect edges, to stationary and mobile-then-stationary behavior in the middle of transects (Fig. 2; Table 3). In contrast, Blanchard's cricket frogs exhibited increased mobility from transect boundary to middle ($\chi^2 = 40.05$, $p < 0.01$), while green tree frogs decreased stationary behavior from transect boundary to middle ($\chi^2 = 353.79$, $p < 0.01$; Fig. 2; Table 3).

Temporally, green tree frogs and Blanchard's cricket frogs exhibited limited seasonal trends in behavior, while southern leopard frogs exhibited a decrease in stationary behavior from June to November (Fig. 3; Table 4). Southern leopard frogs and Blanchard's cricket frogs were more mobile in June (17% and 20%, respectively) and July (23% and 26%,

**Table 3  Post hoc results for Chi squared test of location-specific differences in risk behavior.**

|  | Boundary | Edge | Middle |
|---|---|---|---|
| *L. sphenocephalus* | M & MS > S | MS & S > M | S & MS > M |
| *A. blanchardi* | MS > S > M | MS > M & S | M & MS > S |
| *H. cinerea* | S > MS > M | S > M & MS | ND |

Notes:
M, indicates mobile behavior; MS, indicates mobile-then-stationary behavior; S, indicates stationary behavior, and ND, indicates no difference between behaviors.

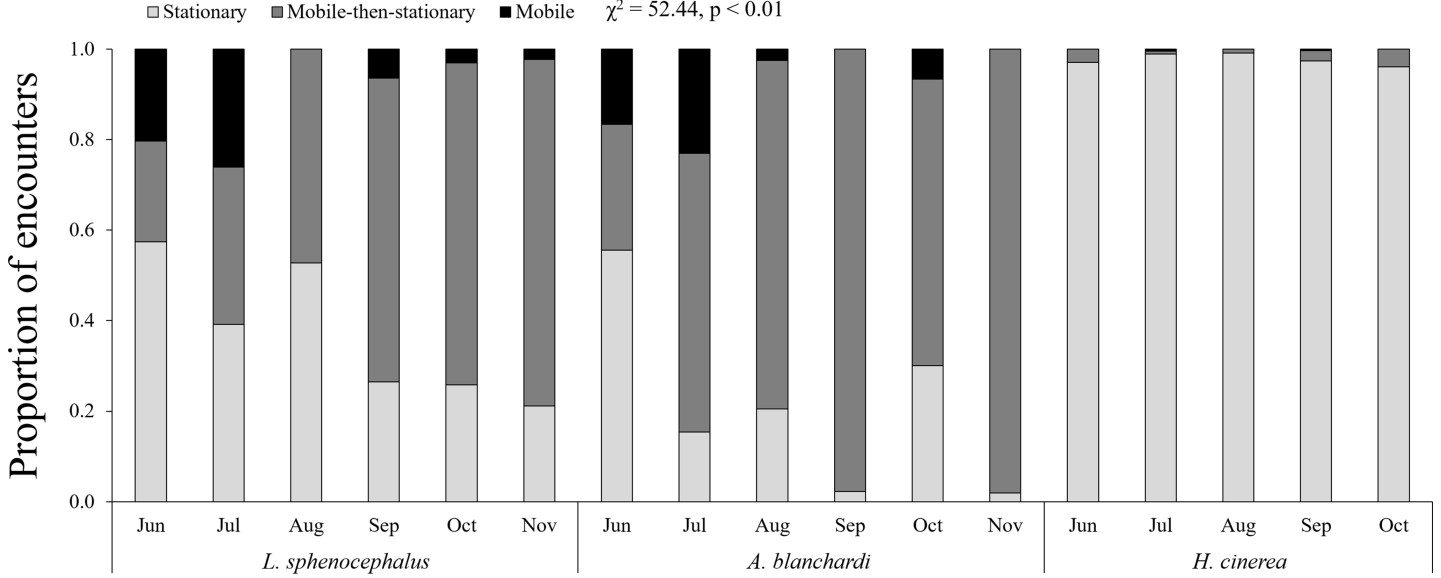

**Figure 3  Monthly patterns in frog behavior.** Monthly patterns in behavior (stationary (S; light gray), mobile-then-stationary (MS; dark gray), and mobile (M; black)) among southern leopard frogs (*L. sphenocephalus*), Blanchard's cricket frogs (*A. blanchardi*), and green tree frogs (*H. cinerea*).

respectively) compared to the rest of the sampling period (1% and 4%, respectively; Fig. 3). All three species exhibited less mobile-then-stationary behavior than expected at the beginning of the sampling period (June and July), and more mobile-then-stationary behavior than expected at the end of the sampling period (October and November; Table 4).

## DISCUSSION

As human impacts continue to alter ecosystems, understanding animal behavior and species abilities to adjust to changing landscapes is of importance for wildlife and habitat management to ensure the maintenance of ecological communities (*Becker et al., 2007*; *Battisti, Poeta & Fanelli, 2016*; *Legrand et al., 2017*; *Pecl et al., 2017*). Our study shows that the risk response of frogs in an altered forest ecosystem is both species- and context-specific, with some species (i.e., southern leopard frogs) at potentially greater risk to human actions in open terrain than others (i.e., green tree frog). As mobile species, frogs are able to avoid predators and other risks through both flight and camouflage (*Marchisin & Anderson, 1978*). However, the efficacy of risk responses may differ based on

**Table 4 Post hoc results for Chi squared test of month-specific differences in risk behavior.**

|                   | June  | July      | August     | September | October | November |
|-------------------|-------|-----------|------------|-----------|---------|----------|
| *L. sphenocephalus* | −MS   | −MS, +M   | −MS, −M    | ND        | +MS     | +MS      |
| *A. blanchardi*     | −MS   | +M        | ND         | +MS       | +MS     | +MS      |
| *H. cinerea*        | ND    | −MS       | ND         | ND        | +MS     | +MS      |

Notes:
M, indicates mobile behavior; MS, indicates mobile-then-stationary behavior, and S, indicates stationary behavior.
A plus (+), indicates a behavior exhibited more frequently than expected; a minus (−), indicates a behavior exhibited less frequently than expected, and ND, indicates no difference between behaviors.

characteristics of habitat (open or covered) and risk (visual, olfactory, heat sensing, non-consumptive, natural vs unnatural; *Gregory, 1979*; *Wells, 2007*; *Bulbert, Page & Bernal, 2015*). Our results suggest that morphology, including the ability to use trees and other vegetation as refuge, and body size can potentially lead to important differences in the response of animals to risk factors.

As predicted, arboreal green tree frogs decreased stationary behavior as the distance to vegetative cover increased, likely due to their propensity to use the canopy of vegetation for refuge (*Dodd, 2013*). Similarly, Blanchard's cricket frogs, a ground-dwelling species, increased mobility as distance to cover increased, which could be attributed to the inherent risk faced by this species based on its small body size (*Werner & Gilliam, 1984*). Supportive of our second hypothesis, southern leopard frogs exhibited the greatest likelihood to freeze in more open terrain (i.e., middle sections of transects), with increased stationary behavior as distance to cover increased. Movement may serve as an indicator to visual predators (*Ioannou & Krause, 2009*), and therefore some ground dwelling frogs may remain stationary, and even flatten themselves against the substrate to avoid detection by predators, such as snakes, birds, and mammals (*Marchisin & Anderson, 1978*). Species-specific differences in risk responses may also be innate—some amphibians actively evade predators (*Tollrian & Harvell, 1999*), some species engage or scare predators (*Altig, 1974*), and many remain motionless and use camouflage to blend in to their environments to avoid detection (*Marchisin & Anderson, 1978*; *Stevens & Merilaita, 2011*).

Yet, species with greater tendencies to remain stationary in open but risky habitats may face greater challenges adjusting to human-altered landscapes, with larger-bodied, ground dwelling animals (e.g., leopard frogs) potentially at greater risk of human-induced injury or mortality in unvegetated habitats compared to tree dwelling species (*Beebee, 2013*; *Sosa & Schalk, 2016*). Camouflage is a primary antipredator response of many frogs (*Marchisin & Anderson, 1978*), and all three study species exhibited stationary behavior (98%, 30%, and 13%, respectively, for green tree frogs, southern leopard frogs, and Blanchard's cricket frogs), overall with more frequent freeze responses in more vegetated transect sections (boundary, edge). In terrestrial habitats, species at risk may be more vigilant as terrain becomes less protected by vegetation or landscape features, and flee into more covered habitat when risk exceeds a threshold in open habitats (*Edut & Eilam, 2003*; *Stankowich, 2008*). Green tree frogs and Blanchard's cricket frogs followed this model,

while southern leopard frogs did not, exhibiting an inverse relationship between distance to cover and flight (*Takada et al., 2018*).

Across human-altered ecosystems, some innate behaviors remain beneficial, and the plasticity of other behaviors make them more advantageous in disturbed ecosystems (*Chace & Walsh, 2006*; *McKinney, 2006*; *McCleery, 2009*). For example, the scavenging nature of some predators increases their foraging efficiency in urban and suburban environments where discarded human food is nutritionally beneficial and widely available, reducing energetic costs (*Fedriani, Fuller & Sauvajot, 2001*; *Merkle et al., 2013*). Similarly, human structures can be readily incorporated and utilized by species that rely on camouflage for protection (*Merilaita, 2003*; *Banos-Villalba, Quevedo & Edelaar, 2018*). However, camouflage in ecosystems used by humans is likely most effective for species that blend-in with elevated structure, both natural and man-made. Indeed, camouflage and freeze responses among ground-dwelling species likely provide little protection from agricultural, landscaping, and transportation practices, each presenting lethal risk (*Trombulak & Frissell, 2000*; *Kirk, Lindsay & Brook, 2011*; *Carvalho et al., 2017*). For many species, humans present unnatural conditions and perturbations that heighten risk (*Cushman, 2006*; *Wong & Candolin, 2015*; *Li et al., 2017*). Yet while camouflage may be the primary risk response of many frogs, this behavior is apparently not ubiquitous among all species, enabling some taxa to thrive in urban and suburban environments (*Rubbo & Kiesecker, 2005*; *Hamer & McDonnell, 2008*; *Scheffers & Paszkowski, 2012*). Identifying species at greater risk and greater adaptability is of importance moving forward in urban and disturbance ecology, as well as conservation, in which identifying general behavioral patterns may be of great value (*Lima & Dill, 1990*; *Wong & Candolin, 2015*; *Battisti, Poeta & Fanelli, 2016*). Our data suggest that arboreal and small-bodied species may be more able to adjust to human impacts than larger-bodied, ground-dwelling species, however, more refined study designs are needed to test this hypothesis.

## Caveats

In light of the observed behavioral patterns and the extensive use of stationary/freeze responses by all three study species, observations may have been biased based on the researcher's ability to detect immobile frogs near or in vegetation (boundary and edge habitats). Yet, substantially more individuals were detected along the boundaries of transect ($n = 2,062$) compared to transect edges ($n = 300$) and middles ($n = 325$). Thus, detection bias was an unlikely factor in shaping the observed trends, and was uniform across all sampling events, because the same observer collected all sampling data.

Seasonal variability in environmental conditions and reproduction often shape animal behavior in subtropical latitudes (*Matich et al., 2017*), and frogs exhibited monthly differences in behavior. Yet, temporal trends in behavior were limited among the study species. Frogs became more mobile from Summer to Autumn, which could be due to decreased vegetation on transects as air temperature decreased, reducing the effectiveness of camouflage, and/or differences in ground and air temperatures, food availability, or breeding cycles (*Saenz et al., 2006*; *Wells, 2007*). Seasonal variability in predation risk could

also lead to seasonal patterns in activity levels and behavior of frogs (*Lode, 2000*; *Sperry et al., 2008*), however, more controlled experiments are needed to test these hypotheses.

## CONCLUSIONS

Habitat degradation continues to pose important conservation concerns across all ecosystems, including forests harvested for timber and converted to agricultural and ranch lands (*Arroyo-Rodriguez et al., 2017*; *Lourenco et al., 2017*; *Tilman et al., 2017*). Amphibians rely on tropical, subtropical, and temperate forests for habitat and food resources, and in strongholds where diseases have not infected populations, habitat management and conservation is of great importance (*Stuart et al., 2004*; *Grant et al., 2016*). Our results suggest that differences in distance to cover, habitat use patterns, and body size may affect frog behavior, with a greater propensity for flight responses to risk by arboreal frogs in open terrain (green tree frogs), and a greater propensity for freeze responses to risk by larger, ground-dwelling frogs (southern leopard frogs). With growing perturbations to many forested regions, these context-specific behaviors are of important consideration for future conservation and management in human-altered systems.

## ACKNOWLEDGEMENTS

We thank the Center for Biological Field Studies and Alan Byboth for providing logistical support during data collection, and the many volunteers that helped survey frogs, including Kaya Moore, Demtri Payblas, Monica Anderson, and Kayla Hankins. We also thank the Texas Research Institute for Environmental Studies for providing logistical support in preparation of the manuscript.

### Funding

The open access publishing fees for this article have been covered by the Texas A&M University Open Access to Knowledge Fund (OAKFund), supported by the University Libraries and the Office of the Vice President for Research, as well as the Office of Research and Sponsored Programs at Stephen F. Austin State University. The funders had no role in study design, data collection and analysis, decision to publish, or preparation of the manuscript.

### Grant Disclosures

The following grant information was disclosed by the authors:
Texas A&M University Open Access to Knowledge Fund (OAKFund).
University Libraries and the Office of the Vice President for Research.
Office of Research and Sponsored Programs at Stephen F. Austin State University.

### Competing Interests

The authors declare that they have no competing interests.

## Author Contributions

- Philip Matich conceived and designed the experiments, performed the experiments, analyzed the data, contributed reagents/materials/analysis tools, prepared figures and/or tables, authored or reviewed drafts of the paper, approved the final draft.
- Christopher M. Schalk authored or reviewed drafts of the paper, approved the final draft.

## Animal Ethics

The following information was supplied relating to ethical approvals (i.e., approving body and any reference numbers):

Research was conducted under Sam Houston State University IACUC #17-02-13-1034-3-01.

## Data Availability

The raw data in the Supplemental Files includes the sampling month, location on transect, study species, and associated risk behavior of frogs.

## Supplemental Information

Supplemental information for this article can be found online at http://dx.doi.org/10.7717/peerj.6956#supplemental-information.

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
