# Peer review of "Move it or lose it: interspecific variation in risk response of pond-breeding anurans"

_PeerJ, doi:10.7717/peerj.6956_

## Round 0.1 · original submission · Major Revisions

In particular, I think you need to present your research question more clearly. You don't give any hypotheses and predictions for examining perceived risk at habitat edges, including potential species specific responses. If your study was designed to be purely exploratory this is also fine, but please just make it clear to the reader.

With respect to body size, it's not clear how many were produced by estimation versus measurement, nor whether there was any calibration or ground truthing so we can know how accurate the estimates were. I think you need to justify this method and/or provide some evidence to show estimated values are useful. If you can't do this, then perhaps this section needs rethinking as it may not be particularly valid.

It would also be really helpful if you could report your statistical analyses more comprehensively. I would like to see the full GLM reported with parameter estimates, and some measure of goodness of fit, and effect size (if possible). I would also suggest testing your model against an intercept-only model to show that the former is informative and helping to explain variation in behaviour.

Reviewer 1 ·

Basic reporting

The authors used professional language throughout

Literature is well referenced

Background and introduction: Much of the introduction is devoted to the impacts of land use change on declining and imperiled species, and the connection to the three common species and their behavioral responses is not clear from a conservation standpoint.
For improvement, the authors should remove unnecessary information (specific examples in general comments) and establish relevant background material that will set up the specific research question.

Figures are of journal quality

Experimental design

This article is within scope of journal

Research question: The research question itself is not clear, and authors provide no predictions, hypotheses, or relevancy for examining perceived risk at habitat edges, including potential species specific responses.

Investigation standard: Estimation of body size in southern leopard frogs not of high standard. Size was either measured or estimated, with no indication of the percentage that were estimated, nor if estimation was in line with actual measurement. This referee suggests removing this section of analysis.

Study design and field methods are sufficiently described, though a figure delineating “middle, edge, and boundary” categories would be helpful to the reader.

While the authors investigated a myriad of factors that might influence movement, there was not sufficient background material addressing the reason for some of the investigations. For instance, change in behavioral choice by season/month was investigated, but there was no explanation or hypothesis laying the framework for this investigation.

Analyses are not clear, and need substantial revision, including reporting degrees of freedom, and some type of goodness of fit for the model. Reporting the GLM (with model fit) and breaking down results by parameter in the model may help reduce the confusion. If the model is informative, showing a test against an intercept only model would help convince readers.

Validity of the findings

Impact and novelty: This study offers an examination of behavioral responses to simulated predation at habitat edges, but does not expand upon the importance of the change in behavioral response to a sufficient degree. While this information may be useful, there needs to be a stronger rational provided.

Data: Beyond the uncertainty of estimating size, data seem adequate, though there is not a clear connection between analyses and results.

Figure one, while only reporting proportions, show a clear biological pattern worth reporting. However, significant revisions of the results are necessary for the validity of the study to be interpretable.

Conclusions: Relevant supporting research is provided, though the organization of the conclusions seems out of place with actual study. As it reads, the first and last paragraph of the discussion are about broad land use change patterns, but this study did not examine broader land use patterns, nor did this study indicate how knowing behavioral response will affect these populations.

Additional comments

By examining the predator response behavior of three anurans at habitat edges, this study offers insight into the potential effects of habitat fragmentation and land use change on three common species of anurans.

Introduction needs to both reduce unnecessary content, and introduce observational study better.

Lines 61-67: Unsure if this section is relevant.

Lines 74-77: Unsure if relevant.

Content setting up the observational study, including the relevance and ecological impact of alterations to predator response need to be added to the introduction. As written, there is little connection to the background and measured responses.

Lines 91-98: Site description unnecessary.

Line 99-101: Why does abundance of an organism make it a good model species?

Lines 99-109, given the scope of your study, it is important to give some background information on each species, especially as it relates to those species being relevant for study. However current background information, especially in lines 105-105 is not useful in the context of this study.

Lines 110-123: Much of this information is relevant to your study, and can be used to make predictions. As such, most of it would better serve the manuscript if moved to the introduction, thus providing the relevant background to set up your research question. Adding in a section on the relevancy and impact of different predator responses would also benefit the introduction.

126: Is a 1.5 meter path cut through a forest enough of a habitat alteration to warrant calling the boundary edge habitat? If so, provide a reference.

129: How is an anthropogenically created habitat edge different from a mown path/forest edge?

145-150: This is more relevant background material that needs to be moved from the methods.

150-151: For what percentage of leopard frogs was SVL measured, and for what percentage was SVL estimated? A breakdown of estimated vs measured sizes needs to be included, in the supplementary material. Even so, it is unclear if these data are accurate enough to include.

157-160: Firstly, why measure month response? There is no relevant background material to support this analysis. Secondly, results should not be reported in the methods.

Analyses and results: The authors state that they utilized a general linear model with logistic regression, though they did not show this model. The authors give specific results, i.e. frogs are more mobile at habitat edges, but do not provide overall responses of species and habitat section.

Discussion: Authors need to provide more ecological context for their results. Lines 215-218 are good, though more synthesis on why understanding behavioral responses to predators will improve future management/conservation work would be beneficial.

Reviewer 2 ·

Basic reporting

The authors need to set up, in the abstract and introduction, that they are comparing different species due in part to their life history characteristics (i.e. arboreal vs semi-aquatic).

The manuscript contains many transitions, that seem to be in there for the sake of transitions. While not disqualifying from a scientific integrity perspective, it can be distracting when reading.

Experimental design

No comment.

Validity of the findings

No Comment.

Additional comments

The manuscript is a valuable addition to the field. It is well-executed. With minor improvements to the writing, it will be suitable for publication.

---

## Round 0.2 · accepted · Accept

Thank you very much for addressing all the reviewers' comments. I'm delighted to now accept your paper for publication.

# Reviewer 1 ·

Basic reporting

no comment

Experimental design

no comment

Validity of the findings

no comment

Additional comments

This manuscript is much improved from the first draft and adequately addressed all comments.